# In-Plane Failure Mechanism and Strength Design of Plate-Tube-Connected Circular Steel Arches

**Xigui Yuan [1], Bo Yuan [2],\*  and Minjie Shi [2]**

1   School of Urban Construction, Chengdu Polytechnic, Chengdu 610218, China
2   Research Center of Space Structures, Guizhou University, Guiyang 550025, China
\*   Correspondence: superyuanbo@163.com

**Abstract:** The in-plane elastoplastic failure mechanism of plate-tube-connected steel circular arches with inverted triangular cross sections is investigated in this study by using theoretical derivation and numerical simulation. First, the in-plane elastic buckling load formula of the arch under full-span uniform radial load (FSURL) is presented. Then, the limited conditions of avoiding the connecting plate and chord local failure before global elastic instability are derived. Lastly, the elastic–plastic failure mechanisms of arches are studied under FSURL, full-span uniform vertical load (FSUVL), and half-span uniform vertical load (HSUVL). It is found that the arch will experience global failure, chord local failure, combined connecting plate and chord failure, and connecting plate local failure under FSUVL and HSUVL. The failure mode is mainly related to the stiffness of the connecting plate. The corresponding design formulas are proposed for the global failure of arches and local failure of the chord. The proposed formulas and FE results are in good agreement.

**Keywords:** plate-tube-connected steel arch; circular arch; global failure; shear failure; in-plane strength design

## 1. Introduction

Arches are widely used in stadium roofs, airports, bridges, and other large-span spatial structures due to their good appearance and bearing capacity. The plate-tube-connected circular steel arch with an inverted triangular cross section is composed of steel plates and tubes. As shown in Figure 1, the axis of the connecting plate is perpendicular to that of the chord, and the connecting plates are distributed uniformly in a radial direction along the axis of the arch at a certain distance. Therefore, this type of arch has good lighting and ventilation performance and can be widely used in building structures. Accordingly, this arch has high research value.

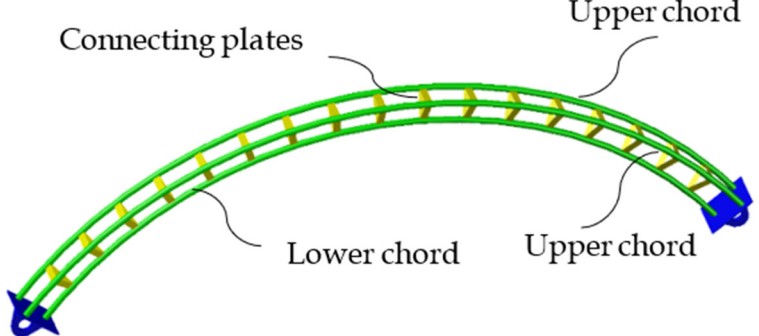

**Figure 1.** Three-dimensional model of plate-tube-connected circular steel arches with inverted triangular cross sections.

The in-plane stability problem is more important than the strength due to the special axis type of arches. As shown in Figure 2, arches can be divided into solid webs and trusses

according to the cross-sectional type of steel arch. The in-plane elastic stability, elastic–plastic failure mechanism, and strength of I-shaped solid-web steel arches were studied in depth by Pi and Trahair [1,2], Pi and Bradford [3], Attard et al. [4], and Zhu et al. [5] by combining theoretical, experimental, and numerical simulation methods. They also proposed the corresponding in-plane strength design formulas of solid-web arches.

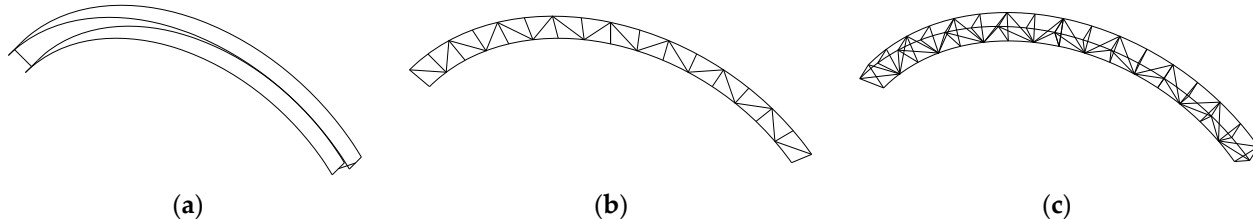

(**a**)　　　　　　　　　　　(**b**)　　　　　　　　　　　(**c**)

**Figure 2.** Cross-sectional types of steel arches: (**a**) solid-web arch; (**b**) plane truss arch; (**c**) spatial truss arch.

Guo et al. [6,7] studied the local failure mechanism of I-shaped solid-web steel arches with sinusoidal wave webs under shear force and proposed the design formula of web shear strength. Guo et al. [8] and Huang et al. [9] numerically studied the elastic stability, elastic-plastic failure mode, and stability of an I-shaped web opening arch. They also proposed the corresponding stability bearing capacity design formula. Guo et al. [10] studied the effect of local buckling of flange and web on the in-plane strength of a box-section arch. Lu et al. [11] experimentally and numerically investigated the in-plane buckling and ultimate resistance of circular steel arches with elastic horizontal and rotational end restraints. A design method for a steel arch with elastic horizontal and rotational end restraints was proposed. Dou et al. [12] numerically studied the in-plane buckling and strength problem of inverted and upright triangular cross-section tabular truss arches with diagonal tubes. They considered the member instability effect on the global stability bearing capacity of the arch and proposed the in-plane strength design formula of a triangular truss arch. Guo et al. [13] studied the stability of a circular arc steel tube truss arch with quadrilateral sections' in-plane stability bearing capacity through experiment and numerical analysis. They also proposed the in-plane stability bearing capacity design formula of circular steel tubular truss arches with a quadrilateral cross section.

However, only a few researchers have studied Vierendeel arch structures. Kinnick [14] studied the effect of shear deformation on the in-plane elastic buckling of double-limb batten lattice arches, and the elastic buckling load formula of double-limb batten lattice arches was derived. It should be noted that its elastoplastic buckling performance has not been studied. Guo et al. [15] studied the in-plane failure mechanism and strength design of circular steel planar tubular Vierendeel truss arches. This provides enlightenment for the research of plate-tube-connected circular steel arches with inverted triangular cross sections in this work.

Compared with the solid-web steel arch, existing research on the stability of the Vierendeel steel tubular truss arch is still relatively lacking, although this structure has been widely used in practical engineering. The plate-tube-connected circular steel arch with an inverted triangular cross section studied in this paper is a new type of Vierendeel truss arch structure; its in-plane elastic stability, elastoplastic failure mechanism, and strength design are rarely reported. Therefore, it is necessary to deeply study the failure mechanism and strength of plate-tube-connected circular steel arches with inverted triangular cross sections to sum up the relevant stability design methods for reference in design and construction.

This work adopts theoretical derivation and a finite element (FE) method to study the in-plane elastic stability and elastoplastic failure mechanism of plate-tube-connected circular steel arches with inverted triangular cross sections. The arch is assumed to be in-plane pin-ended and three kinds of load cases are considered, namely full-span uniform radial load (FSURL), full-span uniform vertical load (FSUVL), and half-span uniform

vertical load (HSUVL). The corresponding design formulas are given for different failure modes to provide a reference for engineering design.

## 2. Finite Element Model

### 2.1. Description of FE Model

The global size and cross-sectional parameters of the arch calculation model are shown in Figure 3, where $L$ is the arch span, $f$ is the arch rise, $R$ is the arch axis radius, $S$ is half of the arch axis length, $H$ is the section height, $L_c$ is the segment length, $L_0$ is the clear spacing of the connecting plates, $B$ is the section width, $D$ is the chord outer diameter, $t_c$ is the chord thickness, $b_w$ is the connecting plate width, and $t_w$ is the connecting plate thickness.

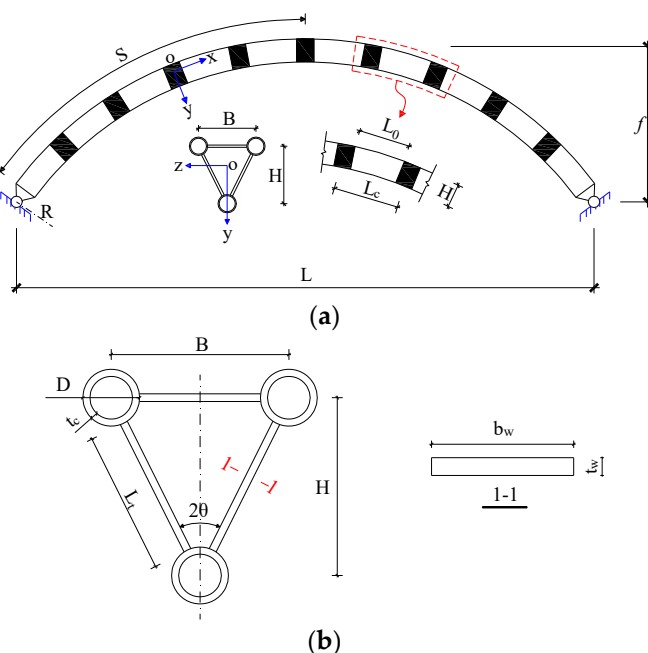

**Figure 3.** Related parameters of the arch: (**a**) global size of the arch; (**b**) cross-sectional parameters of the arch.

In this work, the finite element (FE) analysis software ANSYS is used for numerical simulation, and beam element 188 is utilized to establish the calculation model. Following the method of Guo et al. [15], the plate-tube-connected circular steel arch with inverted triangular cross sections is equivalent to an arch with a rectangular cross section of the same compressive, bending, and shear rigidity along the arch axis. The equivalent model is only used to obtain the elastic internal force and study the deformation of the arch. The other analysis is based on the actual situation to establish a 3D refined beam element model. Three chords are properly stretched out and intersected at the cross-sectional centroid to impose boundary conditions, which are applied at the intersection point. The two ends of the arch foot are assumed to be hinged in plane; thus, only the in-plane rotation angle of the arch foot is released, and the out-of-plane displacement of all chords is restricted. The effects of geometric nonlinearity, material nonlinearity, and geometric initial imperfections on the stability bearing capacity are considered in the elastoplastic analysis of large deflection.

The ideal elastic–plastic material model is adopted in this study. The elastic modulus is $E = 2.06 \times 10^5$ MPa, the Poisson's ratio is $v = 0.3$, the chord's yield strength is $f_{y1} = 235$ MPa, and the connecting plate's yield strength is $f_{y2} = 345$ MPa. The triangular frames at both ends of the model are regarded as rigid members. The first-order antisymmetric buckling mode of the arch is introduced as the initial geometric imperfection into the elastoplastic calculation model with the amplitude $v_0 = S/500$ at a quarter of the developed length, which is consistent with Guo et al. [6,7,15]. This work mainly considers three types of load

cases (Figure 4), namely full-span uniform radial load (FSURL), full-span uniform vertical load (FSUVL), and half-span uniform vertical load (HSUVL).

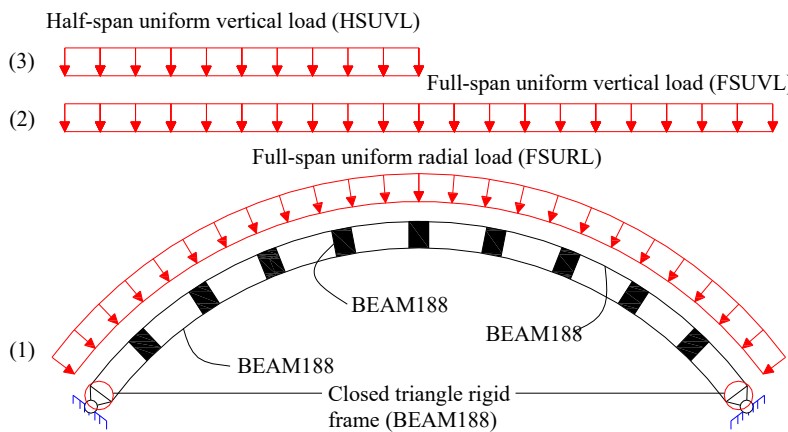

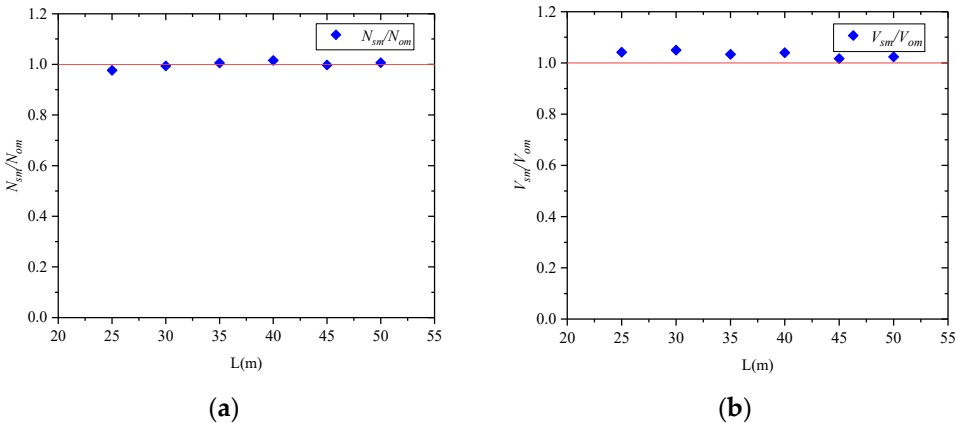

**Figure 4.** Load cases considered in this study.

### 2.2. Verification of Simplified Beam Model

In order to verify whether the simplified beam (sm) model is equivalent to the original model (om), the internal forces of the simplified model and the original at the arch foot are compared. The parameters of the original models are $f/L = 0.3$, $L = 25$–50 m, $B = 0.5$ m, $H = 0.5$ m, $L_0 = H$, $D \times t_c = 0.114 \times 0.01$ m, and $b_w \times t_w = 0.2 \times 0.02$ m. The results are shown in Figure 5, where $N_{sm}$ and $V_{sm}$, respectively, denote the axial force and shear force of the simplified model; $N_{om}$ and $V_{om}$, respectively, denote axial force and shear force of the original model. The results show that the simplified model is equivalent to the original model.

**Figure 5.** Comparison of internal force at arch foot: (**a**) comparison of axial force; (**b**) comparison of shear force.

## 3. In-Plane Elastic Buckling

### 3.1. Section Shear Stiffness

Unlike in the solid-web arch, the effect of section shear deformation on the global elastic buckling load of the arch cannot be neglected. Guo [16] studied the shear stiffness of a steel tubular truss arch with diagonal tubes. Unlike the steel tubular truss arch with diagonal tubes, the plate-tube-connected circular steel arch mainly bears the cross-sectional shear force through the chord. Under pure shear force $V$, if the segment length $L_c$ is small, then the segment can be regarded as straight, and its shear mechanism is equivalent to that of three-limb batten lattice columns. According to the principle of structural symmetry [17], the bending moment distribution of a segment when its reverse bending point is at the midpoint of the chord segment can be easily obtained, as shown in Figure 6. The shear

deformation of the segment under pure shear force is mainly composed of two parts, namely the deformations caused by the bending of the chord, and the connecting plate. Depending on the moment distribution of the segment, the total deformation of the segment under pure shear force *V* is expressed as follows:

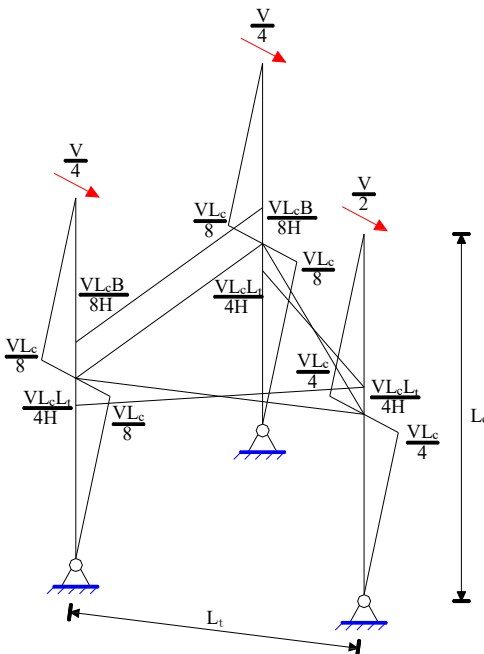

**Figure 6.** Distribution of bending moments of a segment under shear force *V*.

$$\delta = \frac{VL_c^3}{32EI_c} + \frac{VL_c^2L_t^3}{24EI_tH^2} + \frac{VL_c^2B^3}{64EI_tH^2} \tag{1}$$

According to the definition of shear stiffness, we have

$$K_v = \frac{V}{\gamma} = \frac{V}{2\delta/L_c} \tag{2}$$

where $\gamma$ is shear angle. With the introduction of Equation (1) into Equation (2), the expression of section shear stiffness $K_v$ of the plate-tube-connected steel arch with an inverted triangular cross section is obtained as follows:

$$K_v = \frac{1}{\frac{L_c^2}{32EI_c} + \frac{L_cL_t^3}{24EI_tH^2} + \frac{L_cB^3}{64EI_tH^2}} \tag{3}$$

If the shear strain of the connecting plate under shear force *V* is considered in calculating the shear angle $\gamma$, then Equation (3) can be rewritten as Equation (4):

$$K_v = \frac{1}{\frac{L_c^2}{32EI_c} + \frac{L_cL_t^3}{24EI_tH^2} + \frac{L_cB^3}{64EI_tH^2} + \frac{nL_c}{L_tA_tG}} \tag{4}$$

where *n* is the cross-section coefficient, and its value for rectangular cross-section plate is 1.2 according to Timoshenko [18]. $A_t$ is the cross-section area of the connecting plate, *G* is the shear modulus, and $E = 2.6G$.

### 3.2. Buckling Mode of the Arch

Similar to three-limb batten lattice columns, when the chord has a large slenderness ratio, local buckling of the chord will occur before the global buckling of the arch. When the out-of-plane bending stiffness is small, local buckling of the connecting plates will also

occur before global buckling of the arch. The three kinds of buckling modes of the plate-tube-connected arch are shown in Figure 7. The model parameters of different bucklings are shown in Table 1.

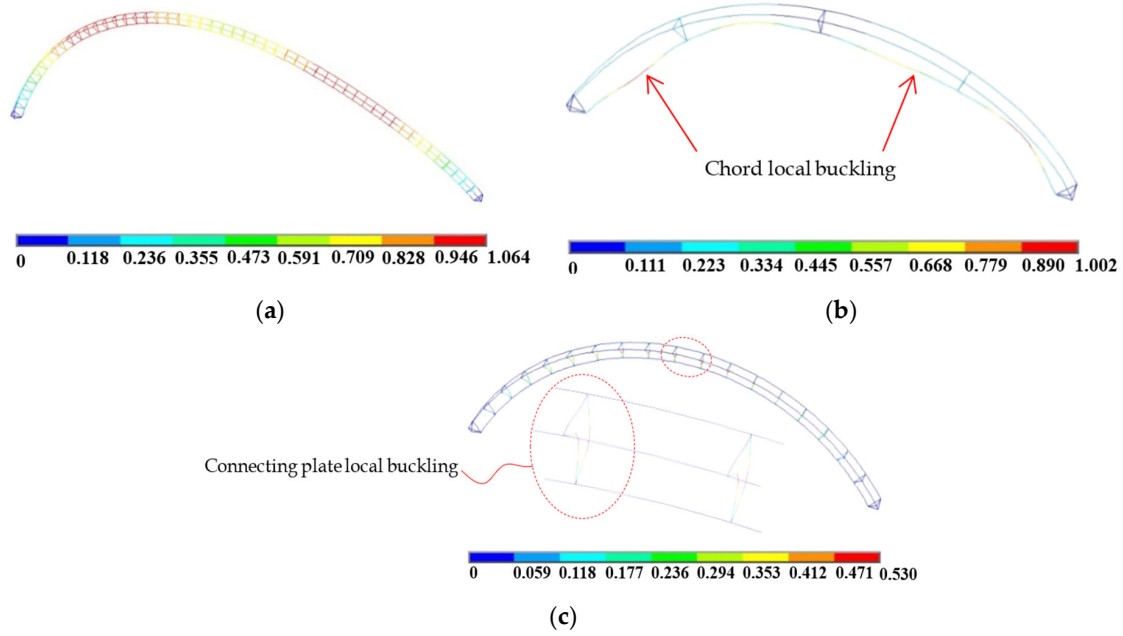

**Figure 7.** Three kinds of buckling modes of the arch: (**a**) global elastic buckling of the arch; (**b**) local chord buckling of the arch; (**c**) local connecting plate buckling of the arch.

**Table 1.** Model parameters of different buckling modes of the arch.

| Buckling Modes | Model Parameters Corresponding to Buckling Modes |
| --- | --- |
| Global buckling | $f/L = 0.3$, $L = 50$, $H = 1$ m, $B = 1$ m, $L_0 = H$, $D \times t_c = 0.114 \times 0.01$ m, $b_w \times t_w = 0.2 \times 0.02$ m |
| Local chord buckling | $f/L = 0.3$, $L = 50$, $H = 1$ m, $B = 1$ m, $L_0 = 2H$, $D \times t_c = 0.114 \times 0.01$ m, $b_w \times t_w = 0.2 \times 0.03$ m |
| Connecting plate buckling | $f/L = 0.3$, $L = 50$, $H = 1.5$ m, $B = 1.5$ m, $L_0 = 2H$, $D \times t_c = 0.114 \times 0.01$ m, $b_w \times t_w = 0.2 \times 0.01$ m |

### 3.3. Global Elastic Buckling Load

Timoshenko [18] derived the in-plane elastic buckling load formula of circular solid-web arches hinged at both ends as follows:

$$q_{cr,0} = \frac{EI_g}{R^3}\left(\frac{4\pi^2}{\Theta^2} - 1\right)$$

(5)

where $EI_g$ denotes the cross-sectional flexural rigidity. However, unlike solid-web arches, the sectional shear deformation on the global elastic buckling load of plate-tube-connected circular steel arches with inverted triangular cross sections cannot be neglected. Kinnick [12] provided the in-plane elastic buckling load formula of double-limb lattice arch hinged at both ends. Guo [19] derived the elastic buckling load formula of steel tubular truss arches with diagonal tubes and inverted triangular cross sections under the effect of cross-sectional shear deformation. According to their results, in this work, the proposed elastic buckling load formula of plate-tube-connected circular steel arches with inverted triangular cross sections under uniform radial load is

$$q_{cr} = \frac{q_{cr,0}}{1 + \frac{q_{cr,0}R}{K_v}}$$

(6)

where $q_{cr,0}$ is the classical elastic buckling load of a pin-ended circular arch under a uniform radial load and is calculated using Equation (5). $K_v$ is the cross-sectional shear stiffness of a plate-tube-connected steel arch with an inverted triangular cross section and is calculated using Equation (4). The results of Equations (5) and (6) are compared with the FE results with different models to verify the rationality of Equation (6) for calculating the in-plane elastic buckling load of a plate-tube-connected circular steel arch with an inverted triangular cross section hinged at the arch foot under FSURL. The parameters include section width $B$, section height $H$, clear spacing of connecting plates $L_0$, rise-to-span ratio $f/L$, and span $L$. The cross-sectional width is uniformly taken as $B = 0.5$ m, the chord size $D \times t_c = 0.114 \times 0.01$ m, and the connecting plate size $b_w \times t_w = 0.2 \times 0.01$ m. The comparison between Equation (6) and FE results is shown in Figure 8.

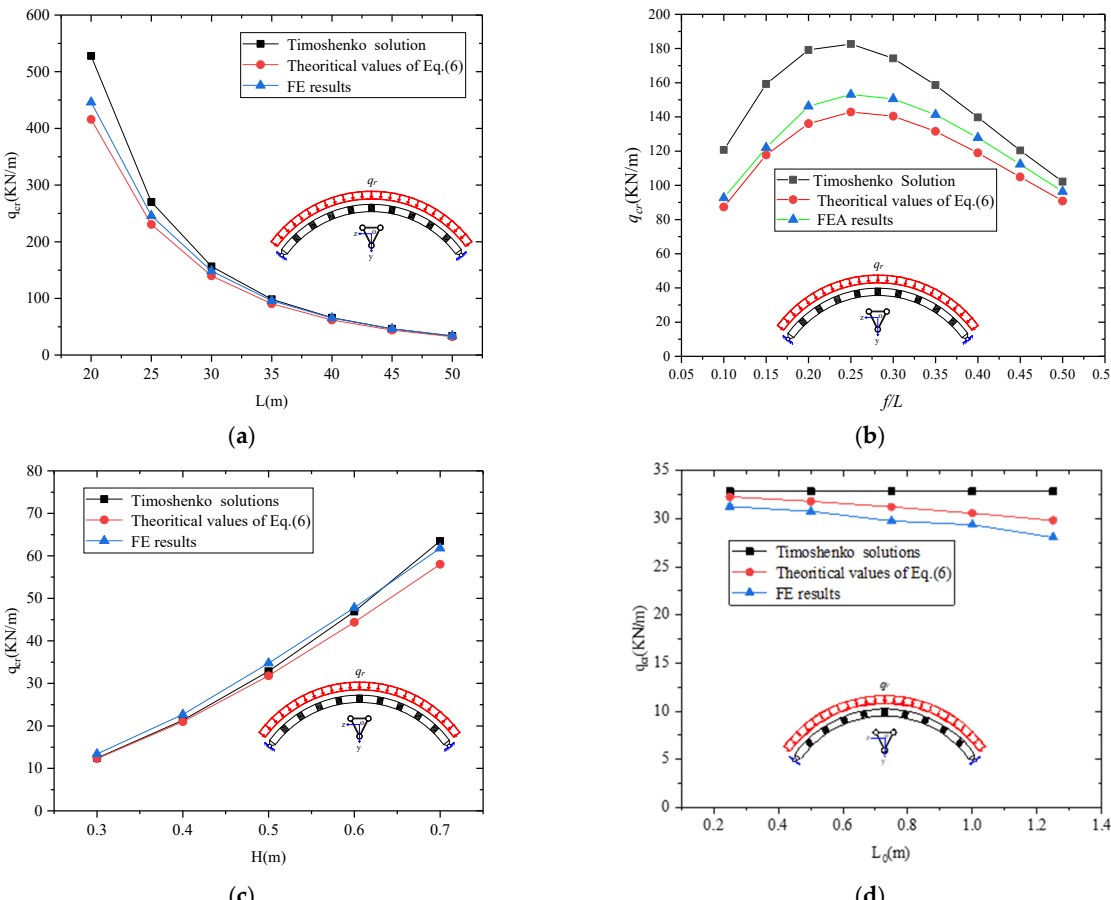

**Figure 8.** Comparison of FE results with the solutions of Equation (6): (**a**) changing span; (**b**) changing rise-to-span ratio; (**c**) changing section height; (**d**) changing clear spacing of connecting plates.

As shown in Figure 8a, the in-plane elastic buckling load of the arch decreases with the increase in span, while the rise-to-span ratio remains unchanged and the span is changed. With the increase in span, the FE results gradually come close to those of Equation (6) and the Timoshenko solution. Specifically, the shear deformation slightly affects the elastic buckling load of the arch as the span increases. Figure 8b shows that when the other geometric parameters of the arch are kept unchanged and only the rise-to-span ratio is changed, the buckling load decreases with the increase in the rise-to-span ratio, then the value decreases after $f/L$ = 0.25. When $f/L = 0.25$, the buckling load is at its maximum. Figure 8c shows that the elastic buckling load of the arch increases with the increase in the section height because the cross-sectional flexural stiffness also increases along with the section height, thereby causing a growth in the global elastic buckling load. Figure 8d shows that the Timoshenko solution will not change when the clear spacing of the connecting plates is changed. The elastic buckling load of the arch decreases with the increase in the clear spacing of the

connecting plates because the shear angle increases with the increase in spacing, thereby decreasing the shear stiffness. Equation (6) shows that the elastic buckling load of the arch also decreases. Figure 8d shows that the elastic buckling load of the arch decreases with the increase in the linear stiffness ratio of the connecting plate to the chord because the decrease in the bending stiffness of the connecting plate will decrease the global bending stiffness of the arch. Accordingly, the elastic buckling load will also increase. The results shown in Figure 8 are in good agreement with the FE results, and the maximum relative error is less than 10%. The results in Figure 8 further show that beam element 188 can be used to establish FE models, and the derivation of section shear stiffness is correct.

### 3.4. Limited Conditions of Member Local Buckling

The plate-tube-connected pin-ended circular steel arch with an inverted triangular cross section will be in pure compression under a uniform radial load. The chord with a large slenderness ratio will buckle before the global elastic buckling of the arch. The out-of-plane elastic buckling of the connecting plate will also occur before the global elastic buckling because the out-of-plane flexural stiffness of the connecting plate is far less than that in the plane. The chord slenderness ratio $\lambda_c$ should be limited to the global slenderness ratio $\lambda_g$, and the slenderness ratio of the connecting plate $\lambda_t$ to the global slenderness ratio $\lambda_g$, to prevent the local buckling of the chord and connecting plate from occurring before the global elastic buckling of the steel arch. The slenderness ratio of the chord is defined as follows:

$$\lambda_c = \frac{L_c}{\sqrt{I_c/A_c}} \tag{7}$$

The global slenderness ratio of the arch is defined as follows:

$$\lambda_g = \frac{S}{\sqrt{I_g/3A_c}} \tag{8}$$

Chord local buckling is a complex problem, and the connecting plates will constrain the end of the chord when this condition occurs. The connecting plate will provide support for the chord. In a large segment, the effect of curvature on the stability of the chord cannot be ignored. Thus, the segment chord can be regarded as circular arches that are elastically embodied at both ends. However, the restraining effect of the connecting plate on the chord will be different with the change in connecting plate size and chord size. This condition brings difficulty in calculating the chord buckling load. Conservatively, the chord is simplified as a circular arch hinged at both ends. The problem of local buckling of the chord is transformed into the problem of solving the critical load of the hinged arch at both ends. The mechanical model of the segment chord is shown in Figure 9.

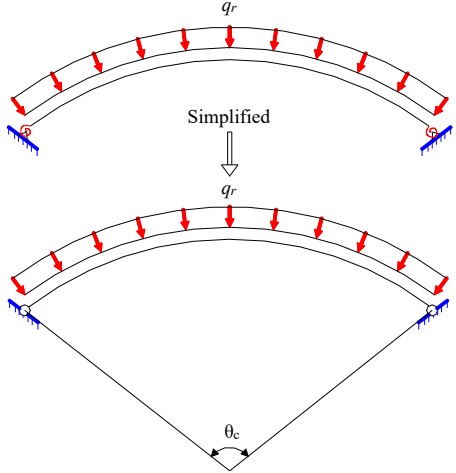

**Figure 9.** Simplified calculation model of the chord.

The elastic buckling load of the tube arch can be calculated using Equation (5) due to the small cross-sectional shear deformation of the steel tube. The elastic buckling load of the upper chord can be written as follows:

$$q_{cr,2} = \frac{EI_c}{\left(R + \frac{H}{3}\right)^3} \left(\frac{4\pi^2}{\theta_c{}^2} - 1\right) \tag{9}$$

where $EI_c$ denotes the chord cross-sectional bending stiffness and $\theta_c$ is the chord segment angle.

Equal stability theory posits that the buckling load of the upper chord should satisfy the following inequality to avoid chord local buckling:

$$N_{cr} = q_{cr}R \le q_{cr,1}\left(R + \frac{H}{3}\right) \tag{10}$$

where $q_{cr}$ denotes the buckling load of plate-tube-connected circular steel arches with inverted triangular cross sections. With Equations (5) and (9) integrated into Equation (10), the inequality can be rewritten as follows:

$$\frac{EI_g}{R^3}\left(\frac{4\pi^2}{\Theta^2} - 1\right)R \le \frac{EI_c}{\left(R + \frac{H}{3}\right)^3}\left(\frac{4\pi^2}{\theta_c{}^2} - 1\right)\left(R + \frac{H}{3}\right) \tag{11}$$

By bringing $\lambda_c$ and $\lambda_g$ into Equation (11) and through simplification, we have

$$\frac{\lambda_c}{\lambda_g} \le \sqrt{\frac{4}{3}\frac{4\pi^2 - \theta_c{}^2}{4\pi^2 - \Theta^2}} \tag{12}$$

Equation (12) shows that when $\theta_c = \Theta$ (that is, no connecting plate is present between the upper and lower chords), $\lambda_c/\lambda_g \le \sqrt{4/3}$, which is the worst case. Usually, $\theta_c < \Theta$, which indicates that $\left(4\pi^2 - \theta_c{}^2/4\pi^2 - \Theta^2\right) < 1$. In summary, the limited condition to avoid chord local buckling is $\lambda_c/\lambda_g \le \sqrt{4/3}$.

The oblique connecting plate of the arch will buckle before the global elastic buckling of the arch occurs. Instability will occur one after another because the transverse connecting plate restrains the inclined connecting plate. No relevant research on the stability of triangular frames is available at present. Thus, a simplified method is used to conservatively estimate the elastic buckling load of the connecting plate. The oblique connecting plate is simplified to an oblique compressive bar hinged at both ends, as shown in Figure 10.

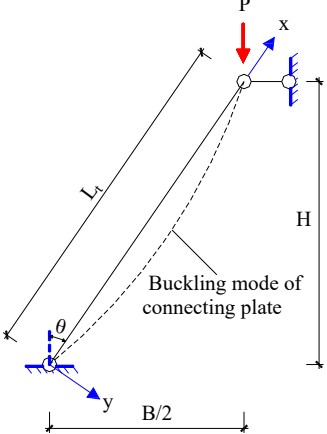

**Figure 10.** Simplified calculation model of the oblique connecting plate.

In accordance with the equilibrium method, the elastic buckling load formula of oblique connecting plates can be obtained as follows:

$$P_{cr} = \frac{\pi^2 E I_t}{L_t^2} cos\theta \tag{13}$$

where $E I_t$ is the cross-sectional flexural rigidity of the oblique connecting plate. Equation (13) can also be rewritten as follows:

$$P_{cr} = \frac{\pi^2 E A_t}{\lambda_t^2} cos\theta \tag{14}$$

where $A_t = b_w \times t_w$, $\lambda_t$ denotes the slenderness of the oblique connecting plate and is defined as follows:

$$\lambda_t = \frac{L_t}{\sqrt{I_t / A_t}} \tag{15}$$

Equal stability theory posits that the following conditions should be satisfied to prevent the oblique local buckling of the connecting plate from occurring before the global elastic buckling of the plate-tube-connected circular arch with an inverted triangular cross section:

$$P_{cr} > N_{cr,0} = q_{cr,0} R \tag{16}$$

Substituting Equations (5) and (15) into Equation (16) yields

$$\frac{\pi^2 E A_t}{\lambda_t^2} cos\theta > \frac{E I_g}{R^3} \left( \frac{4\pi^2}{\Theta^2} - 1 \right) R \tag{17}$$

The limited value of $\lambda_t / \lambda_g$ under FSURL is obtained by bringing $\lambda_g$ and $S = R/2$ into Equation (17).

$$\frac{\lambda_t}{\lambda_g} < \sqrt{\frac{4\pi^2}{4\pi^2 - \Theta^2} \frac{A_t}{3A_c} cos\theta} \tag{18}$$

## 4. In-Plane Failure Mechanism of Arches

### 4.1. Failure Mechanism under FSURL

The plate-tube-connected steel circular arch with an inverted triangular cross section is in a pure compression state under FSURL. The chord may experience elastoplastic buckling before the global elastic–plastic buckling of the arch under the action of axial pressure. Therefore, the slenderness ratio of the chord should be limited to avoid its local elastic–plastic buckling. The Chinese Technical Specification for Steel Tube Structures (CECS280-2010) [20] for the stability of three-limb batten lattice columns specifies that the limb slenderness ratio should meet Equation (19) to avoid local elastoplastic instability.

$$\lambda_c < \min\{0.5\lambda_{0x}, 40\} \tag{19}$$

where $\lambda_{0x}$ is the equivalent slenderness ratio of the arch and is defined as Equation (21).

$$\lambda_{0x} = \sqrt{\lambda_g^2 + \frac{\pi^2}{48}\lambda_c^2(5 + 8\beta_1)} \tag{20}$$

where $\beta_1$ is the ratio of the linear stiffness of the chord to the oblique connecting plate $\beta_1 = i_c / i_t$; $\lambda_g$ is the global slenderness ratio of the arch. $\lambda_c$ is the slenderness ratio of the chord. Equation (18) should be satisfied spontaneously to ensure that the connecting plate does not suffer from local buckling. To investigate the global instability and failure mechanism of a plate-tube-connected circular steel arch with an inverted triangular cross section under FSURL, large deflection elastoplastic analysis is conducted by taking models with rise-to-span ratio $f/L = 0.3$, $L = 20$ m, $B = 0.5$ m, $H = 0.5$ m, $L_0 = H$, $D \times t_c = 0.114$

$\times\ 0.01$ m, and $b_w \times t_w = 0.2 \times 0.02$ m as examples. The load–displacement curve of the vault is drawn in Figure 11.

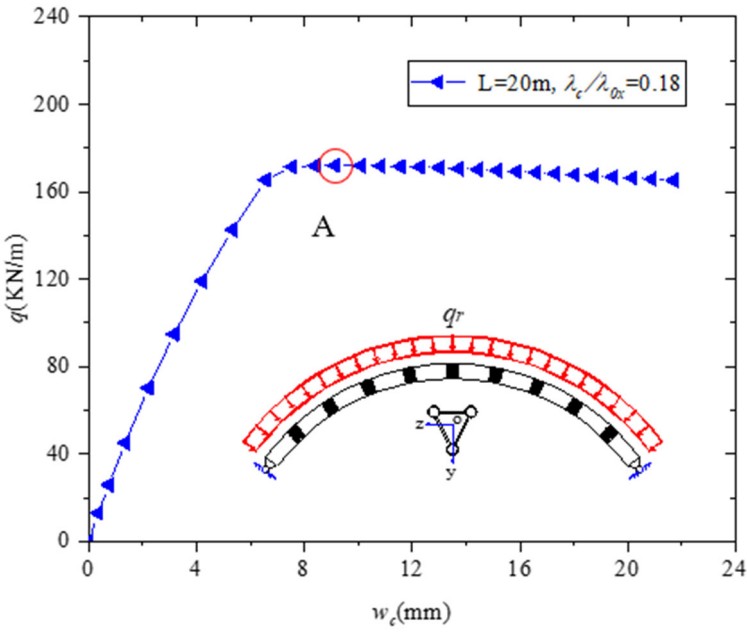

**Figure 11.** Load–displacement curve of the vault under uniform radial load.

Figure 11 shows that the vertical displacement of the vault increases with the increase in the load. The load–displacement curve decreases slowly after the load reaches the ultimate bearing capacity. Therefore, the global elastic–plastic buckling of a plate-tube-connected steel arch belongs to extreme point instability. The first-order antisymmetric deformation is introduced as the initial geometric imperfection. Accordingly, the antisymmetric deformation of the arch occurs with the increase in displacement, and the cross-sectional axial force produces a second-order bending moment. Initially, the chord yields near the quarter point. Then, the plastic region gradually expands. Finally, the whole section yields at the $1/4\,L$ and $3/4\,L$ positions. Figure 12 shows the global stress distribution of the arch at point A. At this point, the connecting plates are still in an elastic state. The large plastic deformation occurs at the $1/4\,L$ position of the lower chord; this condition indicates that the arch has failed.

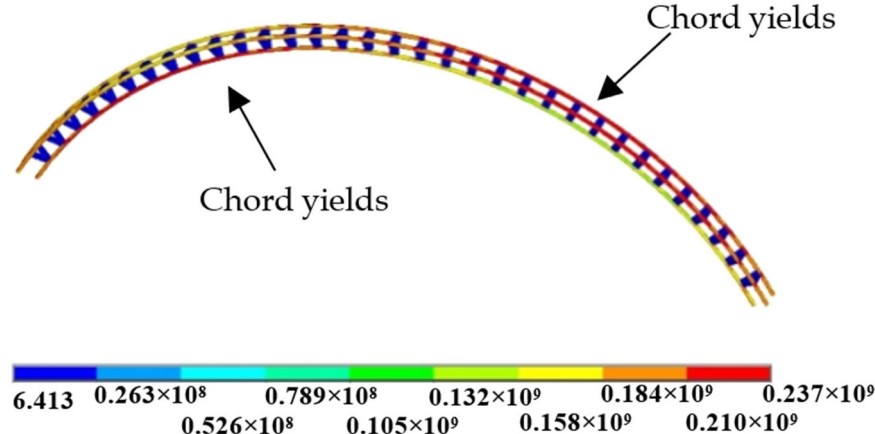

**Figure 12.** Stress distribution of the arch at point A on the curve (Pa).

Pi et al. [21] emphasized that the normalized slenderness ratio can be used to study the stability bearing capacity of the arch under pure compression. The normalized slenderness ratio $\lambda_n$ and reduction factor $\varphi$ are also introduced to study the buckling and yielding of

the plate-tube-connected circular steel arches with inverted triangular cross sections under FSURL. The normalized slenderness ratio $\lambda_n$ is defined as follows:

$$\lambda_n = \sqrt{\frac{N_y}{N_{cr}}} = \sqrt{\frac{3f_{y1}A_c}{q_{cr}R}} \tag{21}$$

The reduction factor $\varphi$ is defined as follows:

$$\varphi = \frac{N_u}{N_y} = \frac{q_u R}{3f_{y1}A_c} \tag{22}$$

where $q_{cr}$ is calculated according to Equation (6) and $q_u$ is the ultimate load under uniform radial load. The parameters of the FE models are $L = 20\text{--}50$ m, $f/L = 0.1\text{--}0.5$, $B = 0.5$ m, $H = 0.5$ m, $b_w \times t_w = 0.2 \times 0.02$ m, and $D \times t_c = 0.114 \times 0.01$ m. The scope of $\lambda_c / \lambda_{0x}$ varies from 0.06 to 0.28. The results of all FE models are plotted in the $\varphi - \lambda_n$ column curve and compared with three types of column curves, namely *a*, *b*, and *c* of GB50017-2017 [22] and Eurocode 3 [23] (Figure 13).

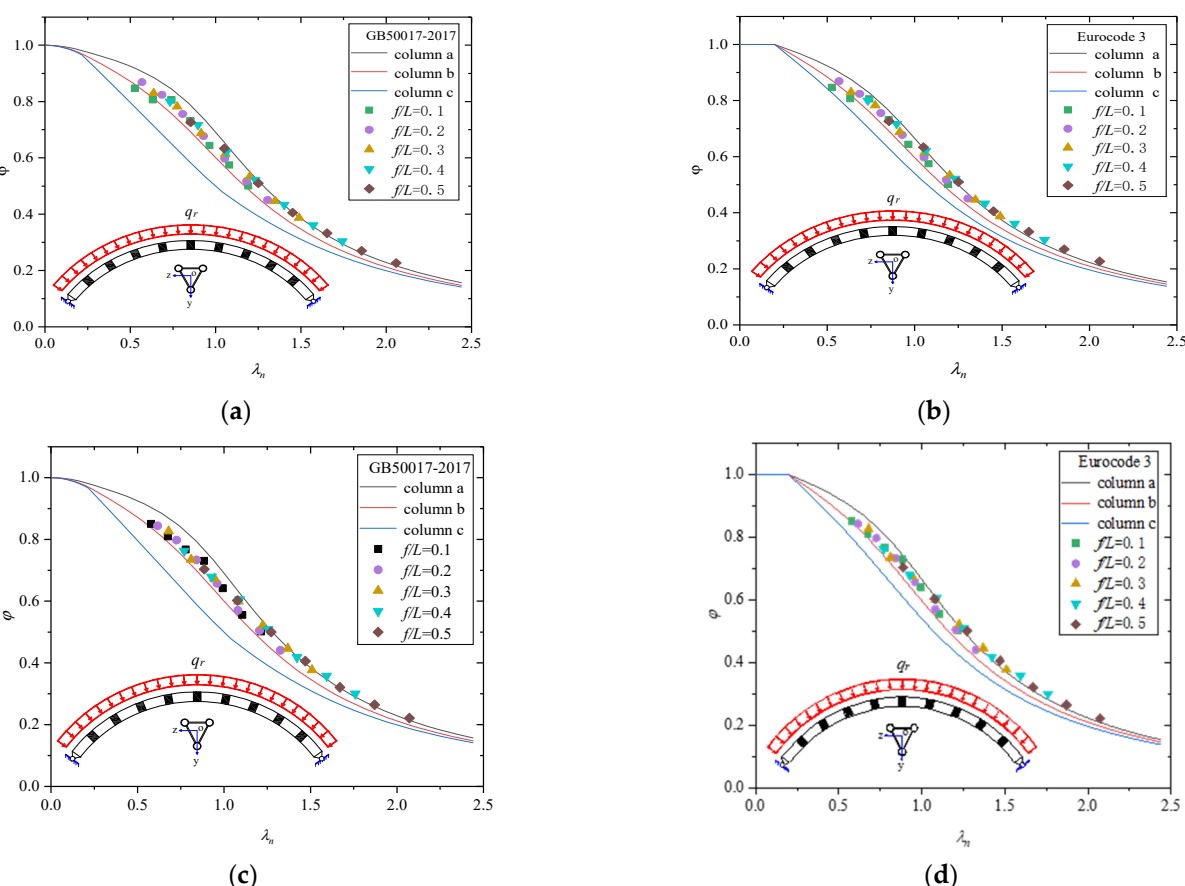

**Figure 13.** $\varphi - \lambda_n$ curve of the arch under FSURL: (**a**) results of FE and GB50017-2017 ($L_0 = H$); (**b**) results of FE and Eurocode 3 ($L_0 = H$); (**c**) results of FE and GB50017-2017 ($L_0 = 1.5H$); (**d**) results of FE and Eurocode 3 ($L_0 = 1.5H$).

Figure 13 shows that column curve *b* of GB50017-2017 or Eurocode 3 can efficiently predict the ultimate bearing capacity of the plate-tube-connected circular steel arch with an inverted triangular cross section under FSURL. Therefore, the stability bearing capacity design formula of the plate-tube-connected circular steel arch under uniform radial load can be written as follows:

$$N_u = \varphi N_y \leq N_y \tag{23}$$

where $\varphi$ is the reduction factor and is taken according to the column curve *b* of GB50017-2017 or Eurocode 3.

### 4.2. Failure Mechanism under FSUVL

The cross-sectional bending moment, shear force, and axial force exist simultaneously under FSUVL. The cross-sectional shear force of Vierendeel structures greatly influences its failure mode. Therefore, the internal force distribution along the axis of the arch should be investigated prior to discussing the failure mechanism of plate-tube-connected circular steel arches with inverted triangular cross sections. A simplified beam model is used to study the internal force distribution of the arch. The corresponding original model parameters are $f/L = 0.3$, $L = 40$ m, $H = 0.5$ m, $L_0 = H$, $D \times t_c = 0.114 \times 0.01$ m, and $b_w \times t_w = 0.2 \times 0.02$ m.

The corresponding internal force distribution is shown in Figure 14, where the maximum internal force is set to 1. Figure 14 indicates that the maximum axial and shear forces are located at the arch foot when the rise-to-span ratio is 0.3. The maximum bending moment is located at the quarter point of the arch. However, the cross-sectional bending moment will be converted into the chord axial. Finally, the cross-sectional shear force will be converted into the chord bending moment. Therefore, the arch may suffer from global and local failure under FSUVL.

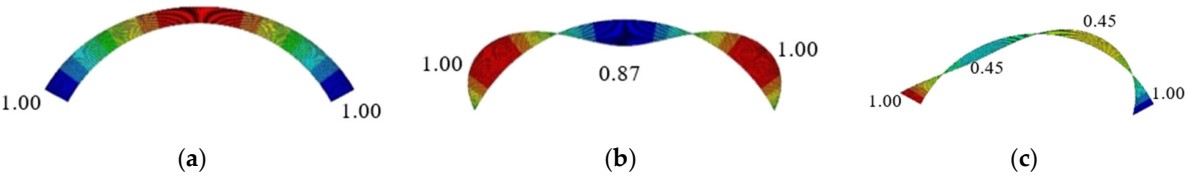

(a)    (b)    (c)

**Figure 14.** Internal force distribution along the arch axis under FSUVL ($\frac{f}{L} = 0.3$): (**a**) axial force; (**b**) bending moment; (**c**) shear force.

### 4.2.1. Global Failure of the Arch

When the segment length of the plate-tube-connected circular steel arch is short, a small bending moment is produced by shear force at both ends of the chord. Meanwhile, the chord axial force produced by the cross-sectional bending moment is large. Thus, global elastic–plastic failure of the arch may occur.

The whole process of large deflection elastic–plastic analysis of the arch is conducted to study the global failure mode of the plate-tube-connected steel arch under FSUVL. The parameters of the arch are $L = 30$ m, $f/L = 0.3$, $B = 0.5$ m, $H = 0.4$ m, $L_0 = H$ and $L_0 = 2H$, $b_w \times t_w = 0.2 \times 0.01$ m, and $D \times t_c = 0.114 \times 0.01$ m. The load–displacement curve of the vault is shown in Figure 15.

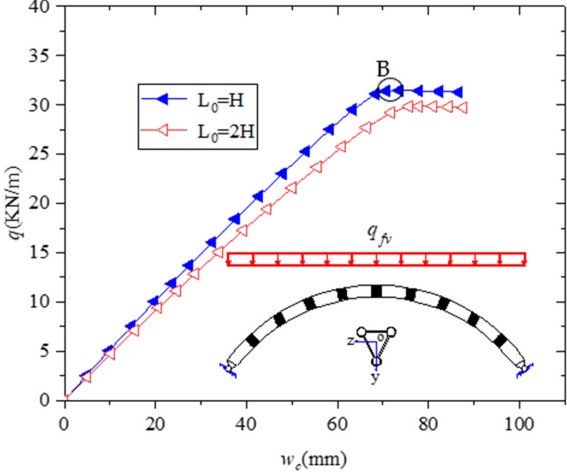

**Figure 15.** Load–displacement curve of the vault under FSUVL.

As shown in Figure 15, the ultimate load of the arch decreases with the increase in $L_c$ when only the segment length $L_c$ is changed. The global stiffness of the arch also changes. As the displacement continues to increase, the plastic area of the lower chord near the arch foot at both ends gradually expands. Figure 16 shows the stress distribution at point B of the load–displacement curve. Because the shear deformation of the cross section will increase the influence of the second-order bending moment, the yield position of the chord is not consistent with the maximum position of the bending moment analyzed by the simplified beam model in Figure 14.

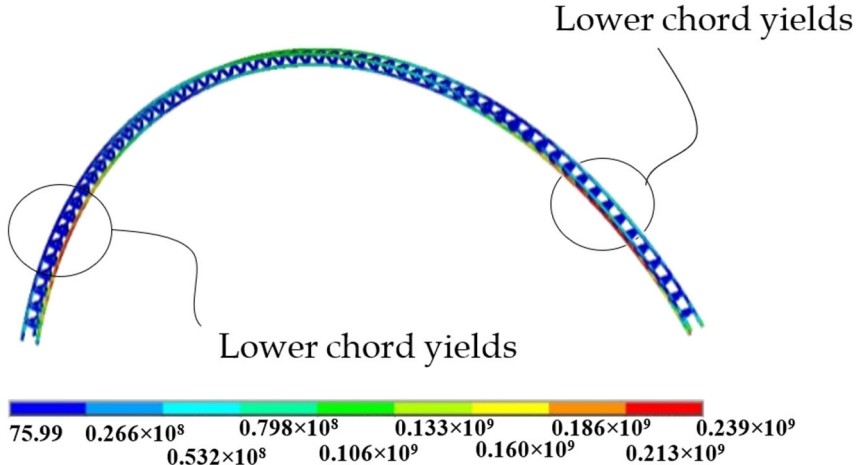

**Figure 16.** Global stress distribution at point B (Pa).

Unlike the solid-web arch, the second-order bending moment caused by the cross-sectional axial force cannot be neglected when the plate-tube-connected circular steel arch undergoes global failure. Thus, the moment amplification factor $\delta$ is introduced to consider the second-order bending moment caused by shear deformation. The global stability bearing capacity design formula for the plate-tube-connected pin-ended circular steel arch with an inverted triangular cross section under FSUVL is proposed as follows [2,15]:

$$\frac{N^*}{\varphi N_y} + \frac{M^*}{M_y} \leq 1 \tag{24}$$

where $N^*$ represents the maximum axial force obtained from the first-order elastic analysis by using the simplified beam model when the arch reaches the ultimate bearing capacity $q_u$; $\varphi$ is the reduction factor, which is taken from the column curve b of GB50017-2017 or Eurocode 3; $N_y$ is the resultant force of the cross-sectional axial force when the chord fully yields and is calculated according to Equation (21); $M_y$ is the cross-section bending moment, $M_y = HN_y/3$; and $M^*$ is defined as follows:

$$M^* = \delta M \tag{25}$$

$M$ is the maximum bending moment obtained from the first-order elastic analysis by using the simplified beam model when the arch reaches the ultimate bearing capacity $q_u$. $\delta$ is the moment amplification factor and is defined as follows:

$$\delta = \frac{1}{1 - \frac{N^*}{q_{cr}R}} \tag{26}$$

where $N^*$ has the same meaning as in Equation (24). $q_{cr}$ is the elastic buckling load of the arch under a uniform radial load and is calculated according to Equation (6).

The FE and Equation (24) results are compared to verify the applicability of adopting Equation (24) to verify the stability of the plate-tube-connected circular steel arch with an inverted triangular cross section under FSUVL. The parameters of the models are $L = 30$–$50$,

$f/L = 0.1\text{–}0.5$, $B = 0.5$ m, $H = 0.4$ m, $b_w \times t_w = 0.2 \times 0.02$ m, and $D \times t_c = 0.114 \times 0.01$ m. The comparison between FE and Equation (24) results is shown in Figure 17.

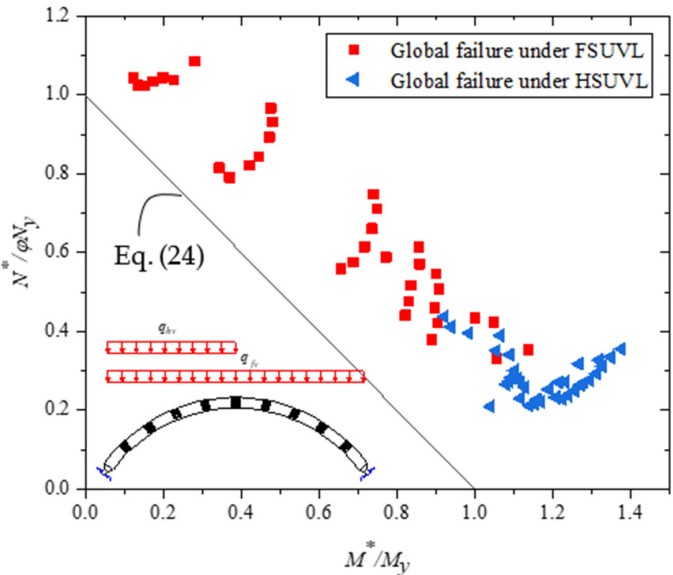

**Figure 17.** Comparison of Equation (24) and FE results under F(H)SUVL.

As observed, all the numerical results are on the right side, the bending moment is the main reason for the global elastic–plastic failure of the plate-tube-connected circular steel arch with an inverted triangular cross section under FSUVL.

### 4.2.2. Local Failure of the Chord

Figure 14 shows that the maximum shear force of pin-ended arches is located at the arch foot. When the segment length $L_c$ of the arch is large, a large chord-end bending moment is generated by the shear force, and chord local failure may occur at the position of maximum shear force. The vault load–displacement curve of the arch with $L = 40$ m, $f/L = 0.3$, $B = 0.9$ m, $H = 0.9$ m, $L_0 = 2H$, $b_w \times t_w = 0.2 \times 0.02$ m, and $D \times t_c = 0.114 \times 0.01$ m is shown in Figure 18 for studying the local chord failure of the arch.

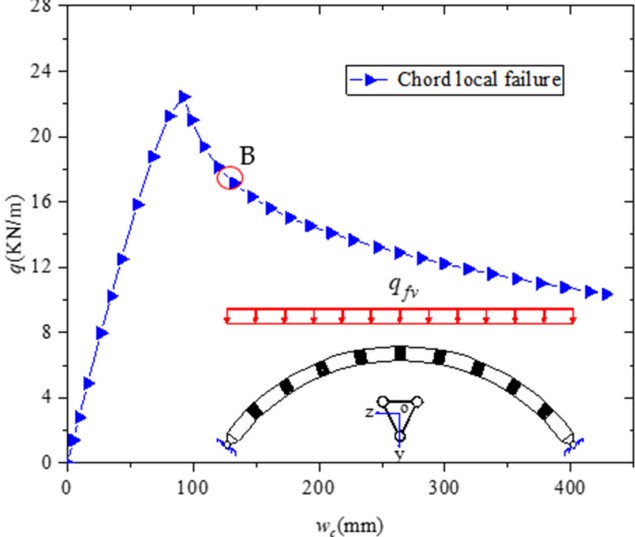

**Figure 18.** The vault load–displacement curve of local chord failure under FSUVL.

Figure 18 shows that, after the steel arch reaches the ultimate bearing capacity, the downward trend of the curve is steep and then tends to be flat after falling within a certain

range. This phenomenon indicates that the arch still has a bearing capacity after reaching the ultimate bearing capacity, because the bending moment produced by shear force will accumulate at the chord end when the stiffness of the connecting plate is high. Finally, the plastic hinges will be formed at the chord end. The plastic hinges at the arch foot of the whole structure can be regarded as the sliding bearing. The whole structure becomes a static one. The arch can continuously bear loads after the chord local failure. The failure mode and global deformation of the arch at point B are shown in Figure 19, and the global stress distribution at point B is shown in Figure 20.

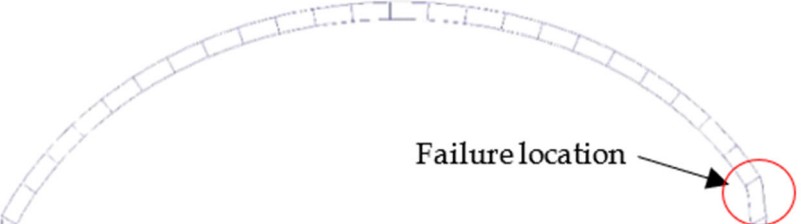

**Figure 19.** Global deformation and failure mode of chord local failure.

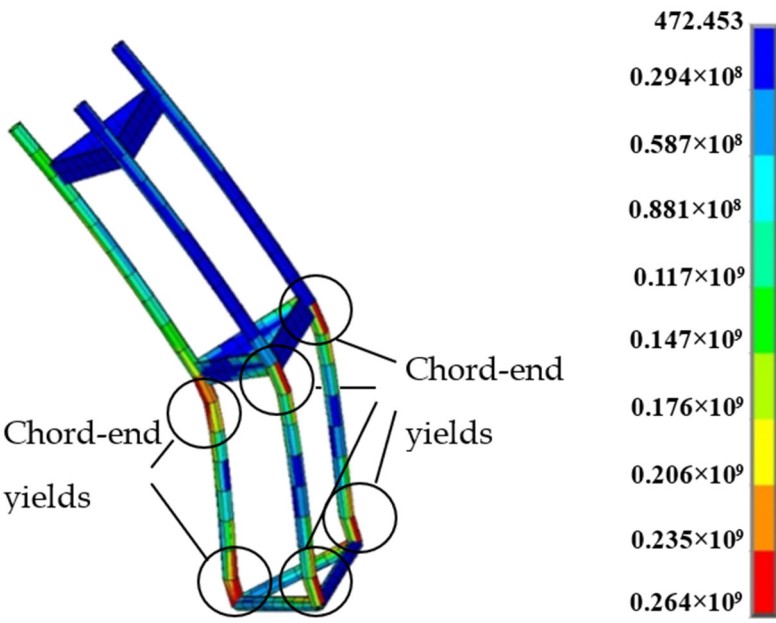

**Figure 20.** Chord local failure global stress distribution (Pa).

### 4.2.3. Combined Connecting Plate and Chord Failure

Figure 6 demonstrates that the connecting plate will resist the sum of the bending moments at both ends of the segment chord. When the connecting plate cannot resist the bending moment, the bending moment will accumulate at the adjacent segment chord end at the arch foot. The structure will eventually fail during the combined chord and connecting plate failure. In the failure process, the parameters of the arch are $L = 40$ m, $f/L = 0.3$, $B = 0.9$ m, $H = 0.9$ m, $L_0 = 2H$, $D \times t_c = 0.114 \times 0.01$ m, and $b_w \times t_w = 0.1 \times 0.02$ m. The load–displacement curve of the vault is plotted in Figure 21. Figure 22 shows the global deformation and failure mode during the combined chord and connecting plate failure at point B. Figures 23 and 24 present the stress distribution at point B.

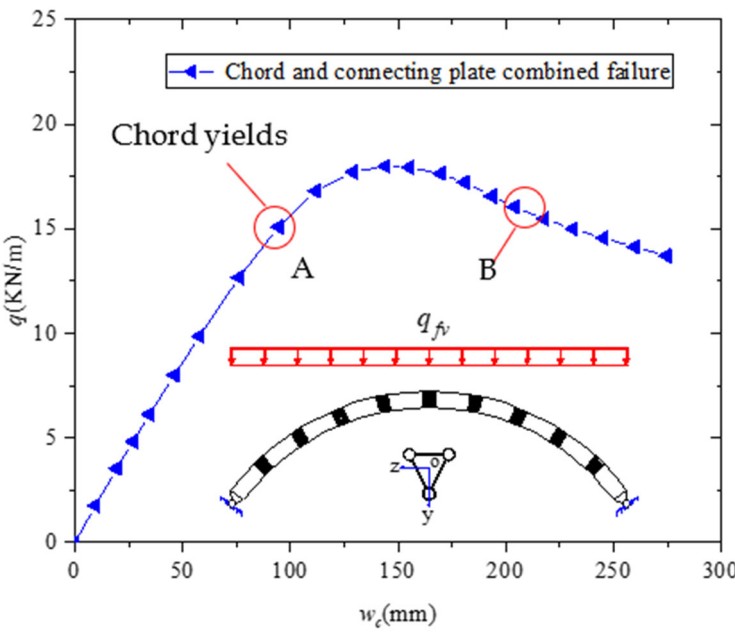

**Figure 21.** Vault load–displacement curve of combined chord and connecting plate failure.

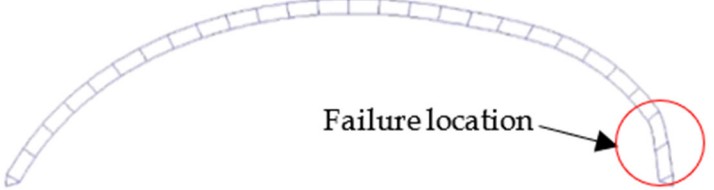

**Figure 22.** Global deformation of combined chord and connecting plate failure.

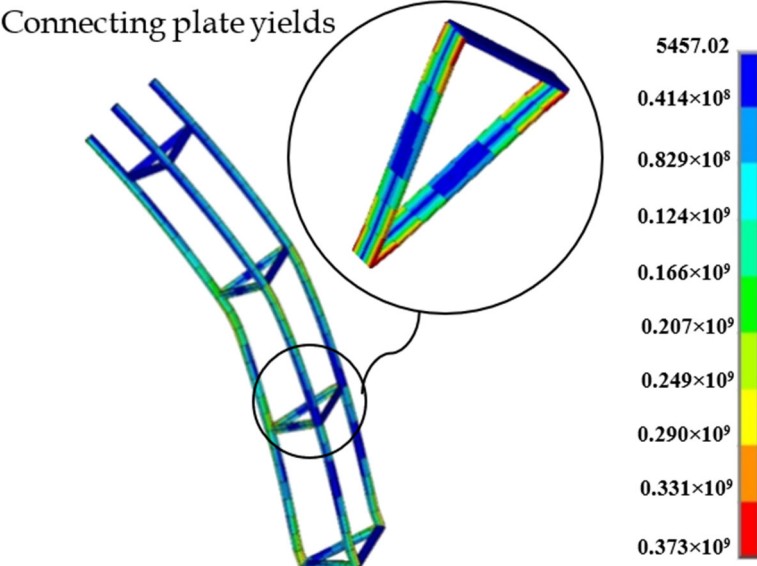

**Figure 23.** Global stress distribution of chord and connecting plate combined failure (Pa).

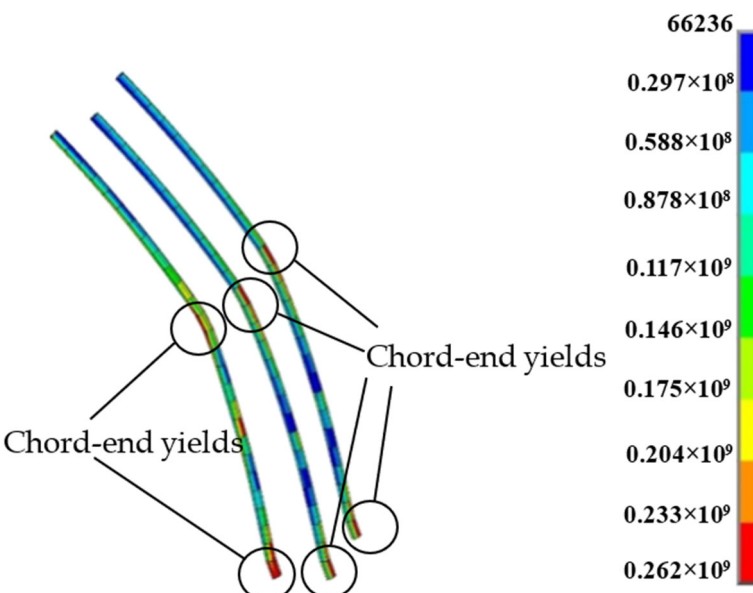

**Figure 24.** Chord stress distribution of chord and connecting plate combined failure (Pa).

The load–displacement curve indicates that the arch can also continue to bear loads when the arch experiences combined chord and connecting plate failure. At point A, the chord yields initially. The connecting plate will fail when it cannot resist the bending moment of the chord end, and the bending moment of the cross section will accumulate at the adjacent segment chord end. Finally, the plastic hinges will form at a further chord end.

### 4.2.4. Connecting Plate Failure

When the flexural rigidity of the connecting plate is gradually reduced, the elastoplastic failure of the connecting plate will occur. No effective connection exists between the upper and lower chords, and their cooperative working ability is poor. In this case, the bearing capacity is weak.

In simulating the failure process, the parameters of the arch are $f/L = 0.3$, $L = 40$ m, $B = 0.9$ m, $H_0 = 0.9$ m, $L_0 = 2H$, $D \times t_c = 0.114 \times 0.01$ m, and $b_w \times t_w = 0.05 \times 0.02$ m. The load–displacement curves of the vault are plotted in Figure 25, and the global deformation and failure mode during connecting plate failure is plotted in Figure 26.

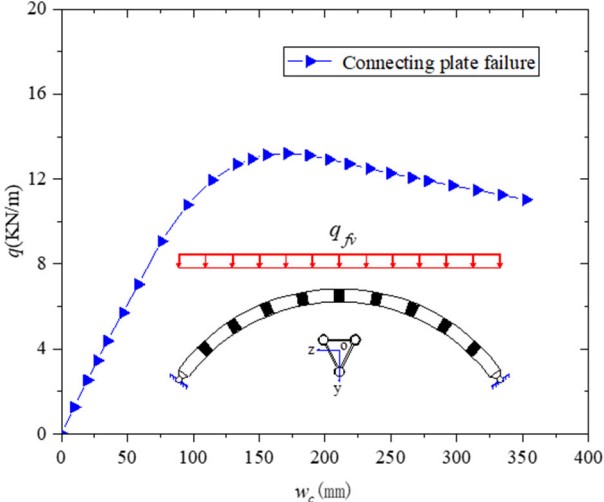

**Figure 25.** Load–displacement curve during connecting plate failure.

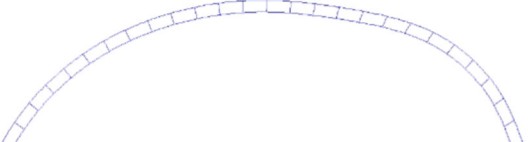

**Figure 26.** Global deformation during connecting plate failure.

Figure 25 shows that the arch can continuously bear loads when the connecting plates fail. However, most connecting plates will yield at both ends with the increase in load due to the small flexural rigidity of the connecting plate in its own plane. After that, the three chords of the arch will continue to carry independently. The final deformation presents global antisymmetric deformation.

4.2.5. Comparison of Three Types of Local Failure

The vault load–displacement curves for three types of local failure modes are plotted in Figure 27 for comparison. The overall stability bearing capacity changes slightly with the change in the connecting plate stiffness when the chord of the plate-tube-connected steel arch with an inverted triangular cross section fails.

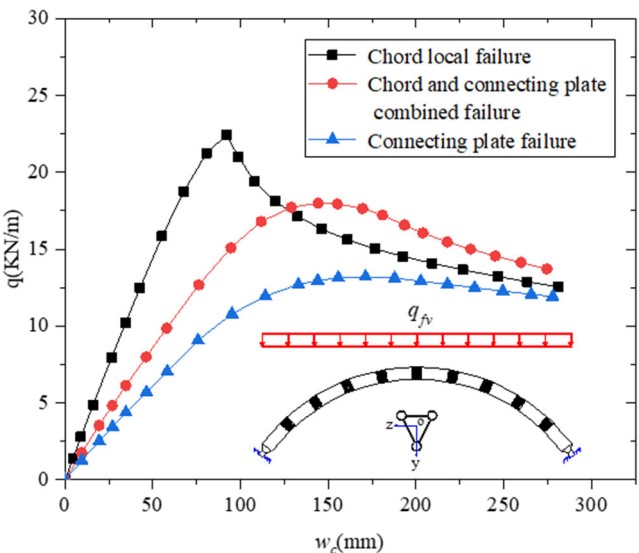

**Figure 27.** Comparison of load–displacement curve for three types of local failure modes.

When the connecting plate stiffness on its own plane decreases, the bearing capacity of the arch decreases gradually, and the failure mode of the plate-tube-connected circular steel arch with an inverted triangular cross section changes from chord local failure to combined chord and connecting plate failure. When the stiffness of the connecting plate is further reduced, the failure mode of the arch changes to connecting plate failure.

In the three types of failure modes, the two previous failure modes are superior to the third one in terms of deformation and bearing capacity. The advantages of chord failure are as follows: (1) The stiffness of the connecting plate is large. The failure and yield of the connecting plate will not occur during the whole loading process when the stiffness ratio of the connecting plate to the chord is constrained. This concept is completely applicable to elastic calculation and does not involve stress redistribution. (2) The fully elastic connecting plate can effectively connect the upper and lower chords. As a result, the overall stiffness and bearing capacity increase. Therefore, chord local failure is preferably designed alone.

Additional FE examples are added to study the relationship between the failure mode and connecting plate size. The parameters of the arch are $f/L = 0.3$, $L = 40$ m, $H_0 = 0.9$ m, and $D \times t_c = 0.114 \times 0.01$ m, and the connecting plate sizes are $b_w \times t_w = 0.2 \times 0.02$ m, $b_w \times t_w = 0.2 \times 0.016$ m, $b_w \times t_w = 0.2 \times 0.014$ m, $b_w \times t_w = 0.15 \times 0.016$ m, $b_w \times t_w =$

$0.15 \times 0.014$ m, $b_w \times t_w = 0.1 \times 0.016$ m, and $b_w \times t_w = 0.1 \times 0.014$ m. The parameters mentioned before are kept unchanged with $L_0 = 2H$ and $L_0 = 2.5H$. The corresponding results are shown in Figure 28.

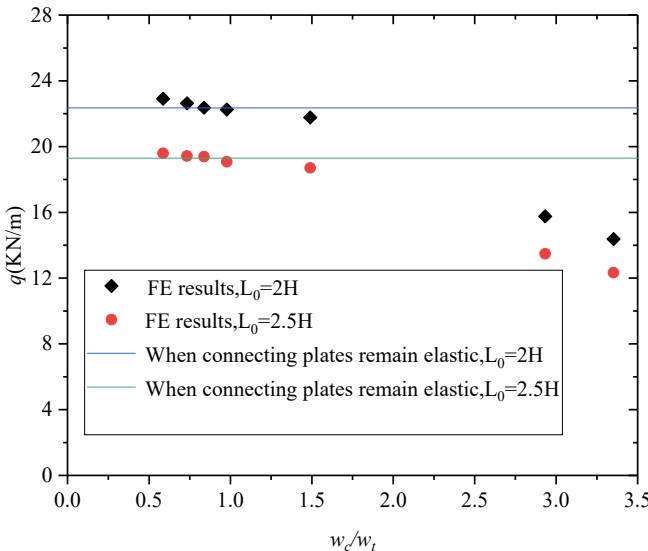

**Figure 28.** Effect of connecting plate size on the ultimate bearing capacity of local failure mode.

When section modulus ratio $W_c/W_t < 0.75$, the connecting plate will remain elastic (where $W_c$ denotes the section modulus of the chord, and $W_t$ denotes the section modulus of the transverse connecting plate). Meanwhile, the connecting plate will be subjected to compression and a bending moment. However, the action of compression is smaller than that of the bending moment. Conservatively, the edge yield criterion is adopted to maintain the elasticity of the connecting plates.

$$\frac{M}{W_n} \leq f_y \tag{27}$$

Integrating the bending moment of the connecting plate end and the chord end into Equation (27) yields

$$\frac{VL_c}{8W_c} \Big/ \frac{VL_c L_t}{4HW_t} \leq \frac{f_{y2}}{f_{y1}} \tag{28}$$

where $W_c$ is the modulus of the chord section, and $W_t$ is the modulus of the connecting plate section. Bringing material parameters into Equation (28) shows that the connecting plate will be elastic when $W_c/W_t < 0.73$.

The plate-tube-connected steel arch with an inverted triangular cross section is mainly supported by chords. The upper two chords bear 1/4 of the shear force $V$. The lower chord bears 1/2 of the shear force $V$. The bending moment of the lower chord is near twice that of the upper chord. The lower chord will first enter the plastic state, but the upper chord will also yield if chord local failure occurs.

Conservatively, referring to GB50017-2017, the edge yield criterion is used to check the upper chord strength at the most disadvantageous position, that is,

$$\frac{N_c}{A_c f_{y1}} + \frac{M_c}{\gamma_x W_c f_{y1}} \leq 1 \tag{29}$$

where $A_c$ and $W_c$ are the area and section modulus of a single chord, respectively; $f_{y1}$ is the yield strength of the chord; and $\gamma_x$ is the plastic development coefficient of the chord section. In the circular pipe section, $\gamma_x = 1.15$, and $N_c$ is defined as follows:

$$N_c = \frac{N_{max}}{3} + \frac{\delta M_{max}}{2H} \tag{30}$$

$M_c$ is defined as follows:

$$M_c = \frac{V_{max}L_c}{8} \tag{31}$$

where $V_{max}$ is the maximum shear force obtained from the first-order elastic analysis by using the simplified beam model when the arch reaches the ultimate bearing capacity $q_u$; $N_{max}$ and $M_{max}$ denote the maximum shear force and bending moment within the segment where the shear force is maximum, respectively; similarly, the values of $N_{max}$ and $M_{max}$ are also calculated by a simplified model when the arch reaches the ultimate bearing capacity $q_u$. $\delta$ is the moment amplification factor, $\delta < 1.4$.

Equation (29) is validated by the FE method. The arches with a rise-to-span ratio of $f/L = 0.15$–$0.5$, $L = 30$–$60$ m, $B = 0.9$ m, $H = 0.9$ m, $L_0 = 2H$, $b_w \times t_w = 0.3 \times 0.02$ m, and $D \times t_c = 0.114 \times 0.01$ m are considered. The ultimate bearing capacity of all plate-tube-connected steel arches with inverted triangular cross sections under chord failure is calculated according to Equation (29) and drawn in Figure 29.

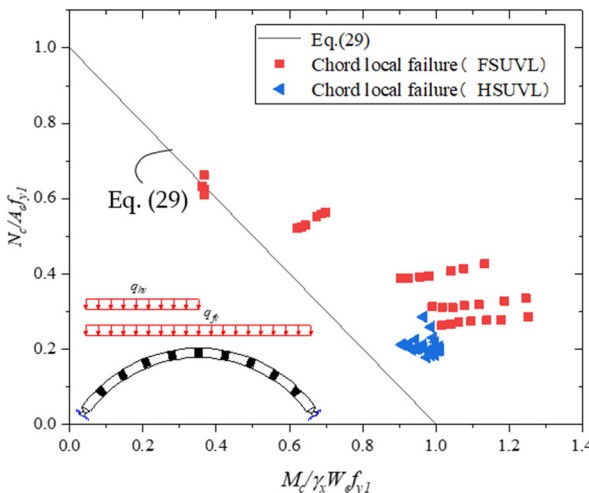

**Figure 29.** Comparison of Equation (29) with FE results.

Figure 29 shows that the data points deviate to the right side, thereby indicating that the local chord failure is mainly caused by the bending moment. All data points are above Equation (30), which can be used to estimate the local failure strength of the plate-tube-connected steel arch with an inverted triangular cross section under the full-span uniformly distributed vertical load.

### 4.3. Failure Mechanism under HSUVL

Under HSUVL, the plate-tube-connected circular steel arch with an inverted triangular cross section is in a state of compression and bending. The distribution law of the internal force along the axis is calculated by the simplified beam model, as shown in Figure 30. The corresponding refined model parameters are $f/L = 0.3$, $L = 40$ m, $H = 0.4$ m, $L_0 = H$, $D \times t_c = 0.114 \times 0.01$ m, and $b_w \times t_w = 0.2 \times 0.02$ m. The numerical value in Figure 30 indicates the relative value of internal force. The maximum value is set to 1.

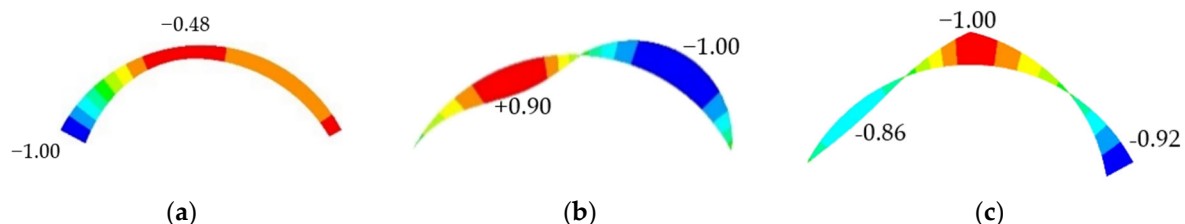

**Figure 30.** Distribution law of the internal force of arch under HSUVL: (**a**) axial force; (**b**) bending moment; (**c**) shear force.

Figure 30 shows that the maximum shear force of the arch under HSUVL is located in the midspan. The maximum axial force is located at the arch foot. The maximum bending moment is located at 1/4 and 3/4 spans. The distribution characteristics of the internal force along the axis manifest that the plate-tube-connected steel arch will have the same elastoplastic and local failure modes of the arches under FSUVL.

4.3.1. Global Failure Mechanism under HSUVL

The parameters are $f/L = 0.3$, $L = 20$ m, $B = 0.5$ m, $H = 0.4$ m, $D \times t_c = 0.14 \times 0.01$ m, $b_w \times t_w = 0.2 \times 0.02$ m, and $L_0 = 2H$. The overall stress distribution when the arch reaches the stable ultimate bearing capacity is shown in Figure 31 for large deflection elastic–plastic analysis. Figure 31 shows that the lower chord will yield at the 1/4 and 3/4 span positions of the arch, and the arch will undergo global antisymmetric deformation.

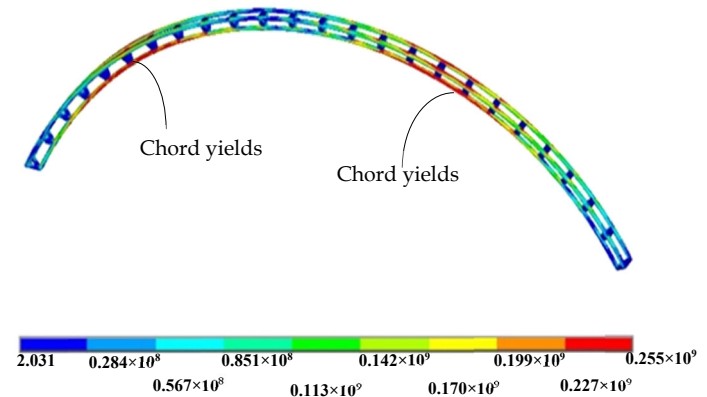

**Figure 31.** Global stress distribution of global failure under HSUVL (Pa).

In the supplementary example, the FE model parameters are $f/L = 0.15$–$0.5$, $L = 20$–$50$ m, $B = 0.5$ m, $H = 0.4$ m, $L_0 = H$, and $b_w \times t_w = 0.2 \times 0.02$ m to verify if Equation (24) can be used to check whether the strength of the overall failure of the arch under HSUVL is suitable. Figure 17 presents the calculation result, which indicates that adopting Equation (24) to verify the stability bearing capacity is safe.

4.3.2. Local Failure Mechanism under HSUVL

The parameters of the unified arching are $f/L = 0.3$, $L = 35$ m, $B = 0.9$ m, $H = 0.9$ m, $L_0 = 2H$, $D \times t_c = 0.14 \times 0.01$ m, and $b_w \times t_w = 0.2 \times 0.03$ m in studying the chord local failure of the plate-tube-connected circular steel arch with an inverted triangular cross section under HSUVL. Figure 32 plots the load–displacement curve at the vault. Figure 33 presents the global deformation and failure mode at point A. Figure 34 show the global stress distribution and chord stress distribution.

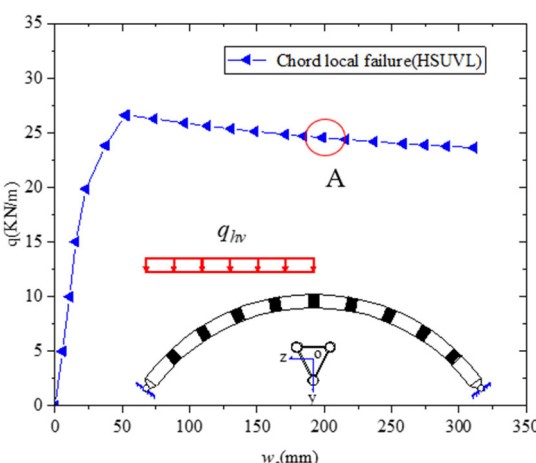

**Figure 32.** Load–displacement curve of chord local failure under HSUVL.

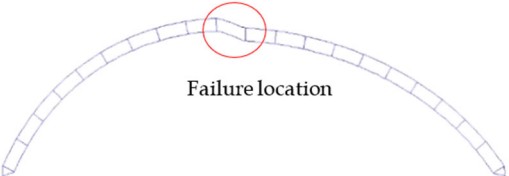

**Figure 33.** Global deformation and failure mode at point A during chord local failure under HSUVL.

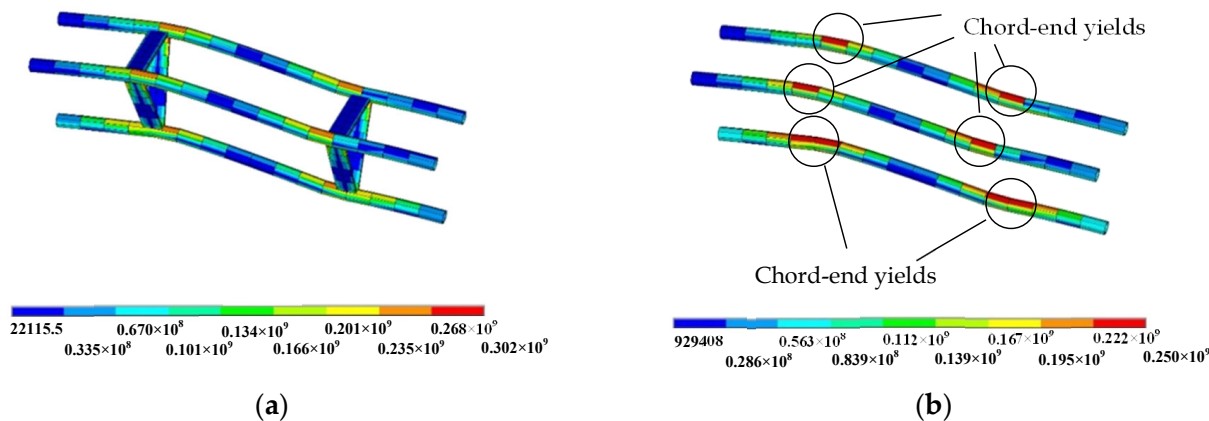

**Figure 34.** Stress distribution during chord local failure (HSUVL): (**a**) global stress distribution at point A(Pa); (**b**) chord stress distribution at point A(Pa).

The comparison result of Figures 32–34 indicate that the arch can still bear loads continuously when the arch reaches the stable ultimate bearing capacity because the plastic hinge will form when the chord end yields, and the chord in mid-span is similar to sliding bearings. The whole structure will then become static, and can thus bear loads continuously.

In studying the two other types of local failure progress of the arch under HSUVL, the parameters of the models are $f/L = 0.3$, $L = 35$ m, $B = 0.9$ m, $H = 0.9$ m, $L_0 = 2H$, $D \times t_c = 0.14 \times 0.01$ m, and connecting plate sizes $b_w \times t_w = 0.15 \times 0.03$ m and $b_w \times t_w = 0.05 \times 0.03$ m. A full progress analysis is conducted. The load–displacement curves of the two types of local failure are shown in Figure 35. The failure mode and global deformation of the combined chord and connecting plate failure and connecting plate failure are shown in Figure 36.

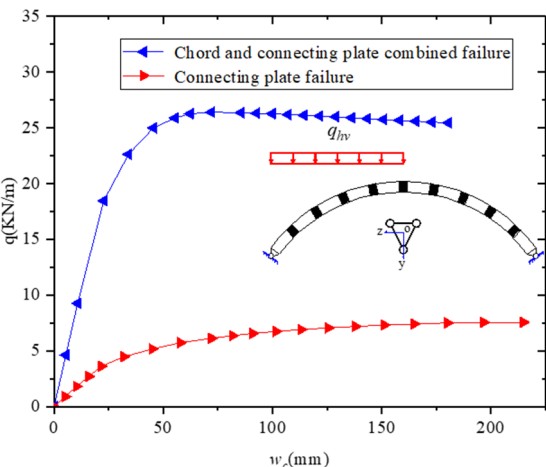

**Figure 35.** Load–displacement curves of chord and connecting plate failure under HSUVL.

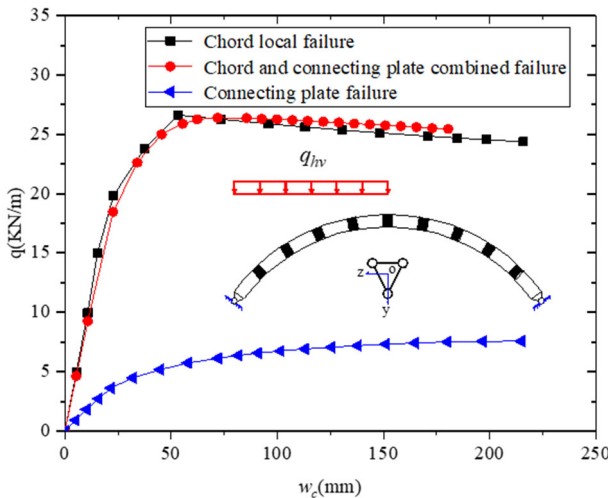

<div style="text-align:center">(a)</div>

<div style="text-align:center">(b)</div>

**Figure 36.** Global deformation and failure mode of the arch under HSUVL: (**a**) combined chord and connecting plate failure; (**b**) connecting plate failure.

Figure 35 shows that the arch will not lose its bearing capacity when the two other local failure modes occur. Arches will present global antisymmetric deformation with the increase in vault vertical displacement. The load–displacement curves of three types of local failure modes are plotted in Figure 37 for analysis.

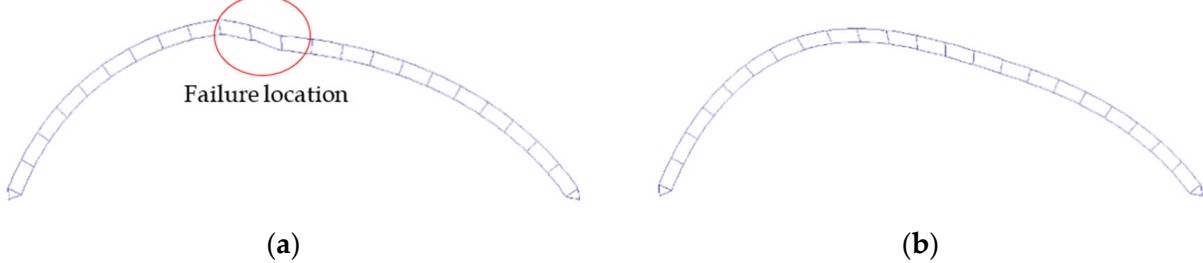

**Figure 37.** Load–displacement curves for three types of local failure modes under HSUVL.

Figure 37 demonstrates that when local failure occurs under HSUVL, the bearing capacity of local chord failure is better than that of the two other types of local failure mode. Similarly, local chord failure is chosen as the allowed local failure mode. The results of FE and Equation (29) are compared in Figure 29 to verify whether Equation (29) is

suitable for checking the strength of the arch during chord local failure. The parameters of the FE models are $f/L = 0.1\text{–}0.5$, $L = 20\text{–}60$ m, $B = 0.9$ m, $H = 0.9$ m, $L_0 = 2H$, $b_w \times t_w = 0.4 \times 0.03$ m, and $D \times t_c = 0.14 \times 0.01$ m.

Figure 29 shows that Equation (29) can predict the strength of local chord failure under HSUVL. The numerical points are on the right of the curve, thereby indicating that the bending moment plays a major role during chord local failure.

## 5. Design Recommendations

When the plate-tube steel circular arch with an inverted triangular cross section experiences local failure, the bearing capacity will not be lost immediately. The bearing capacity of the chord will also be higher than that of the two other local failure modes. Accordingly, the chord local failure can be selected as the control design. The section modulus ratio of the connecting plate and the chord should be guaranteed $\frac{W_c}{W_t} < 0.7$ to ensure that the connecting plate is in an elastic state when the chord is in chord local failure. The parameters of the arch can be determined according to the actual ventilation and lighting requirements. Equations (23) and (24) are used to verify the in-plane global stability when the arch is under pure compression and compression bending, respectively. Moreover, when the chord local failure is allowed, the design Equation (29) can be used to verify the strength of the arch.

## 6. Conclusions

In this work, the in-plane elastic buckling of plate-tube-connected steel circular arches with inverted triangular cross sections under FSURL is discussed. The failure mechanisms under FSURL, FSUVL, and HSUVL are studied comprehensively. The following conclusions are obtained.

- The limiting conditions to avoid local failure of structural members are discussed. When the slenderness ratio of the chord to the arch under FSURL satisfies $\lambda_c/\lambda_g < \sqrt{4/3}$, the chord can be prevented from becoming unstable before the global buckling of the arch occurs. When the slenderness ratio satisfies Equation (18), the connecting plate can be prevented from buckling before the global elastic buckling of the arch occurs. The formula for calculating the shear stiffness of the cross section of the arch is derived theoretically and combined with Equation (6), and the elastic buckling load of the arch when it is subjected to global antisymmetric elastic buckling can be calculated.

- The global elastic–plastic failure of the arch will occur under FSURL, FSUVL, and HSUVL. Under FSURL, the arch is subjected to uniform compression and will yield at 1/4 of the span and the upper chord at 3/4. The ultimate bearing capacity under pure compression can be verified by Equation (23). The reduction factor $\varphi$ is calculated according to column curve *b* of GB50017-2017 or Eurocode 3. When the arch undergoes global elastoplastic failure under FSUVL, it will yield at the lower chord of both arch feet. Meanwhile, the lower chord yields near 1/4 L and 3/4 L under HSUVL. The proposed Equation (24) can be performed to check the global stability of the arch under the action of FSUVL or HSUVL.

- Under FSUVL and HSUVL, three types of local failure modes of the plate-tube-connected steel circular arch with an inverted triangular cross section are observed. The main influencing factor in the local failure mode of the arch is the change in the stiffness of the connecting plate. A comparison of the three failure modes shows that the chord failure mode is better than the two other types. Therefore, the design of the arch can be controlled by the local failure of the chord. The proposed Equation (29) is used to check the strength of the chord local failure, and the ratio of $\frac{W_c}{W_t}$ should be less than 0.7 to ensure that the connecting plate remains in an elastic state.

**Author Contributions:** Conceptualization, X.Y. and B.Y.; methodology, X.Y. and B.Y.; software, X.Y. and M.S.; validation, X.Y., B.Y. and M.S.; formal analysis, X.Y.; investigation, X.Y.; resources, B.Y.; data curation, X.Y. and B.Y.; writing—original draft preparation, X.Y.; writing—review and editing, X.Y., B.Y. and M.S.; visualization, X.Y. and M.S.; supervision, M.S.; project administration, B.Y.; funding acquisition, X.Y. and B.Y. All authors have read and agreed to the published version of the manuscript.

**Funding:** National Natural Science Foundation of China (Grant 51168010).

**Data Availability Statement:** The datasets generated and/or analyzed during the current study are available from the corresponding author upon reasonable request.

**Acknowledgments:** This study was supported by the National Natural Science Foundation of China (Grant 51168010). Their financial support is gratefully acknowledged.

**Conflicts of Interest:** The authors declare no conflict of interest.

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
