# Peer review of "In-Plane Failure Mechanism and Strength Design of Plate-Tube-Connected Circular Steel Arches"

_buildings, doi:10.3390/buildings13040956_

Round 1

Reviewer 1 Report

Report on the manuscript

Title:  In-plane failure mechanism and strength design of plate-tube-  connected circular steel arches

Authors: Xigui Yuana, Bo Yuan, Minjie Shi

Journal: Buildings

Manuscript number: buildings-2274135

The in-plane elastoplastic failure mechanism of plate-tube-connected steel circular arches with inverted triangular section is investigated in this study by using theoretical derivation and numerical simulation. The failure mode is mainly related to the stiffness of the connecting plate. The corresponding design formula is proposed for the global and chord local failure. The proposed formula and FE results are in good agreement.

The results of the study are of low level but they may be of interest for the researcher.  The author has done an extensive bibliographic research identifying the problems determined by this study. The authors proved a good capacity for synthesis, summarizing the research results, formulating conclusions of the research carried out and defining future research directions. Through a meticulous and systematic approach, the authors went through the specific stages of applied scientific research. The analytical and graphic parts of the thesis intertwine and complement each other organically, offering a unitary, modern and interdisciplinary character in a field of great interest. The way of illustrating the results is clear, the language is concise. Academic speaking, the paper is well written, the material is judiciously divided and organized and correct from scientific point of view. Some changes are, however, necessary. For these reasons I can recommend the acceptance of this paper after some corrections presenteReport on the manuscript

Title:  In-plane failure mechanism and strength design of plate-tube-  connected circular steel arches

Authors: Xigui Yuana, Bo Yuan, Minjie Shi

Journal: Buildings

Manuscript number: buildings-2274135

The in-plane elastoplastic failure mechanism of plate-tube-connected steel circular arches with inverted triangular section is investigated in this study by using theoretical derivation and numerical simulation. The failure mode is mainly related to the stiffness of the connecting plate. The corresponding design formula is proposed for the global and chord local failure. The proposed formula and FE results are in good agreement.

The results of the study are of low level but they may be of interest for the researcher.  The author has done an extensive bibliographic research identifying the problems determined by this study. The authors proved a good capacity for synthesis, summarizing the research results, formulating conclusions of the research carried out and defining future research directions. Through a meticulous and systematic approach, the authors went through the specific stages of applied scientific research. The analytical and graphic parts of the thesis intertwine and complement each other organically, offering a unitary, modern and interdisciplinary character in a field of great interest. The way of illustrating the results is clear, the language is concise. Academic speaking, the paper is well written, the material is judiciously divided and organized and correct from scientific point of view. Some changes are, however, necessary. For these reasons I can recommend the acceptance of this paper after some corrections presented in the attached file.

Reviewer 2 Report

I have attached my comments in the file. 

Besides, I would like to know whether there is a confirmation and comparative result between the theoretical results and experimental results by you or others. It may increase the strength of the proposed equations. 

Round 2

Reviewer 1 Report

Thanks to the authors for this new improved version.